# Validation of a French version of the Breakthrough Pain Assessment Tool in cancer patients: Factorial structure, reliability and responsiveness

Elise Perceau-Chambard[1]*, Sylvain Roche[2,3,4,5], Colombe Tricou[1], Catherine Mercier[2,3,4,5], Cécile Barbaret[6], Andrew Davies[7], Katherine Webber[7], Marilène Filbet[1], Guillaume Pierre Serge Economos[1]

1 Palliative Care Center, Hôpital Lyon Sud, Hospices Civils de Lyon, Lyon, France, 2 Service de Biostatistique-Bioinformatique, Hospices Civils de Lyon, Pôle Santé Publique, Lyon, France, 3 Laboratoire de Biométrie et Biologie Évolutive, Équipe Biostatistique-Santé, Villeurbanne, France, 4 Université de Lyon, Lyon, France, 5 Université Lyon 1, Villeurbanne, France, 6 Palliative Care Unit, Centre Hospitalier Universitaire de Grenoble, La Tronche, France, 7 Imperial College London, London, United Kingdom

* Elise.perceau-chambard@chu-lyon.fr

**Data Availability Statement:** The supporting data are available in the Supporting information.

## Abstract

### Objective

Breakthrough cancer pain should be properly assessed for better-personalized treatment plan. The Breakthrough Pain Assessment Tool is a 14-item tool validated in English developed for this purpose; no French version is currently available and validated. This study aimed to translate it in French and assess the psychometric properties of a French version of the Breakthrough Pain Assessment Tool (BAT-FR).

### Methods

First, translation and cross-cultural adaptation of the 14 items (9 ordinal and 5 nominal) of the original BAT tool in French language was made. Second, assessments of validity (convergent, divergent and discriminant validity), factorial structure (exploratory factor analysis) and test-retest reliability of the 9 ordinal items were done with data of 130 adult cancer patients suffering from breakthrough pain in a hospital-academic palliative care center. Test-retest reliability and responsiveness of total and dimension scores derived from these 9 items were also assessed. Acceptability of the 14 items was also assessed on the 130 patients.

### Results

The 14 items had good content and face validity. Convergent and divergent validity, discriminant validity and test-retest reliability of the ordinal items were acceptable. Test-retest reliability and responsiveness of total and dimensions derived from ordinal items were also acceptable. The factorial structure of the ordinal items had two dimensions similar to the original version: "1—pain severity and impact" and "2—pain duration and medication". Items

**Funding:** The Hospices Civils de Lyon "Young Researchers Grant" financially supported this work.

**Competing interests:** Andrew Davies and Katherin Webber developed the original English version of the tool.

2 and 8 had a low contribution to the dimension 1 they were assigned and item 14 clearly changed of dimension compared with the original tool. The acceptability of the 14 items was good.

## Conclusion

The BAT-FR has shown acceptable validity, reliability and responsiveness supporting its use for assessing breakthrough cancer pain in French-speaking populations. Its structure needs nevertheless further confirmation.

## Introduction

Cancer pain is one of the most prevalent cancer related symptom [1, 2], resulting in disturbing patient's quality of life and functioning, and increasing the burden of symptoms belonging to the same cluster, such as fatigue, insomnia or mood disturbance [3–5]. Cancer pain can be characterized by the different types of pain (somatic nociceptive pain, visceral nociceptive pain, or neuropathic pain) [6], but also by its manifestation as persistent, transient, or described as a breakthrough cancer pain. Breakthrough cancer pain is a transient exacerbation of pain that occurs despite relatively stable controlled background pain which might be triggered by a predictable specific event or an unpredictable one [7]. It affects 40 to 80% of cancer patients [8], depending on cancer site, stage, treatments and intercurrent diseases. It has various characteristics, which are different between individuals and vary over time for a same individual. However, common features includes a quick onset up to three minutes, short duration up to thirty minutes and a moderate to severe intensity [9, 10].

A prerequisite for improving patients' quality of life through effective pain management is proper pain assessment [11], it will guide the personalized pain medication plan [6]. Considering its brief and sudden characteristics, breakthrough cancer pain is not properly assessed using generic pain assessment tools. This might prevent clinicians to adequately identify it, and chose specific medications such as intranasal fentanyl sprays or sublingual fentanyl pills [6, 12].

In the purpose of better assessing breakthrough pain, Webber et al. have developed and validated the Breakthrough Pain Assessment Tool (BAT) [13]. This tool is a fourteen item tool of which nine are related to pain and five to its treatments [13], which is now widely recognized as a valid and reliable tool for use in clinical settings. It has been translated in various languages [14, 15], but not into French, preventing French languages countries to use it while French is the fifth most spoken language worldwide.

Our study aimed to translate the BAT into French and assess the psychometric properties (validity, reliability and responsiveness) of a French version of the Breakthrough Pain Assessment Tool (BAT-FR).

## Material and methods

### The Breakthrough Pain Assessment Tool (BAT)

The BAT is a 14 question tool designed to clinically assess breakthrough pain in cancer patients [13]. The tool includes nine questions related to pain and five related to its management. The first question uses a body shape to locate the painful areas. The next questions uses free text (four questions), 10-points rating scales (six questions), and categorical scales (three questions).

### Translation and adaptation of the original tool

We used a process of translation-adaptation adapted from the one proposed by the World Health Organization [16]. First, a French native speaker made a forward translation laying on keeping the units of meaning (rather than an exact literal translation) from the original English version to a French one.

Second, a review board of pain experts reviewed the French version for identification of discrepancies or inadequate translations. Third, an English native speaker backward translated the French version into an English one. It was then, reviewed again by the review board to assess its equivalence with the original BAT. There was no need to modify the previous French version.

Finally, ten cancer patients suffering from pain were asked to assess the readability, and understandability of the tool. Slight changes have been made after this step. The BAT-FR is provided as S1 File.

This entire process with experts and a small sample of patients gave evidence for good content and face validity. The overall process was co-supervised by the original developers of the instrument who are co-authors of this article.

### Study settings and participants

Data collection ran from March 2015 to September 2019 in the Lyon Sud University Hospital Palliative Care Center.

**Participants and setting.** The study was conducted in the Lyon's University Hospital—France. The inclusion settings were a University hospital palliative care center providing palliative care services for more than 1000 patients yearly.

All patients visiting the palliative care center were consecutively screened for inclusion. Inclusion criteria were: being over 18 years old, having cancer pain, having a regular opioid treatment for a background cancer-related pain for at least one week, and having been diagnosed from breakthrough cancer-pain by a specialist cancer pain physician. Inclusion criteria also included being a fluent French speaker, able to understand written French, to complete the questionnaire and to sign informed consent. Non-inclusion criteria were already known cognitive impairment (when listed in the medical history of the patient), and overwhelming fatigue.

1268 patients were screened for inclusion.

Exclusion criteria were the absence of pain (597), refusing to take part (20), agony (207), tiredness (13), inability to consent (235), trouble in understanding French (14), having already participated (23). Seven were excluded for an unknown reason. It resulted in one-hundred and thirty one included patients who received a questionnaire.

**Procedure and data collection.** Participants had to fill the study questionnaire on inclusion, 24h after and 7 days after (noted as inclusion, visit 2 and 3 in the following). The test conditions were the same at each measurement. The questionnaire had two parts, the first recorded socio-demographic data (gender, age, cancer type and localizations, metastatic status, and Performance Status rated using the five-point Eastern Cooperative Oncology Group (ECOG) tool [17], the second was made of the BAT-FR (described above) and a validated French version of the Brief Pain Inventory (BPI) [18, 19].

The BPI is an instrument designed to assess the severity of pain and the functional impairment due to pain. The instrument includes one categorical scale question, one body shape for the localization of pain, four 10-points Likert-scales for rating pain intensity at various times, one free-text question and seven numerical scales to assess the impact of pain on the patient's daily life.

## Statistics

**Sample size.** As for the English version, the nine ordinal items were used to obtain scores and the five nominal items gave additional information and help to interpret the scores. To determine the sample size for factor analysis, we used the rule of 5 x number of parameters [20]. This rule gave 130 patients because the model of the confirmatory factor analysis on the nine ordinal had 26 parameters (with the same structure as the English structure and items with 11 response categories considered as continuous). Moreover, this sample size was greater than the number of parameters for the exploratory factor analysis (up to 36 for four dimensions). According to COSMIN guidelines, a sample size of 130 was considered adequate to assess the other properties of the questionnaire [21].

**Demographic and clinical characteristics of the sample.** Continuous variables were described using the median, the quartiles and the range values (and mean±SD for age and delay). Categorical variables were described using the number and percentage of patients in each category.

**Acceptability of the BAT-FR.** Acceptability was assessed using the percentage of missing data for each of the 14 items at each assessment. A cut-off of 5% of missing data was chosen for the definition of non-acceptability [13].

**Factorial structure of the BAT-FR.** As for the original English version, factor analysis was on the nine ordinal items: n°2, 5, 6, 7, 8, 9, 11, 12 and 14 (see Weber 2014 [13], Table 3). The five other items were nominal (free text or no order in the response categories) and thus could not be subject to a factor analysis; they gave additional information for the clinicians to interpret the ordinal items.

To assess if factor analysis is relevant, a test similar to the Bartlett's sphericity test was used: the Chi-Square Test of Model Fit for the Baseline Model of uncorrelated dependent variables, available in Mplus for model fit information (p-value $\leq 0.05$ indicates that a factor analysis may be useful with the data) [22].

To assess if the structure of the BAT-FR is the same as that of the English version [13], a confirmatory factor analysis (CFA) was conducted on the scores of the nine ordinal items on inclusion. It showed that the model of the English version (two dimensions and each of the nine ordinal items reflecting only the dimension it was supposed to measure) did not fit to data of BAT-FR (results not shown).

Therefore, an exploratory factor analysis (EFA) has been done on the nine ordinal items to produce hypotheses on (i) the number of dimensions and (ii) the structure of the items for the retained dimensions [23]. As some items are ordinal, the WLSMV (Weighted Least Squares Mean and Variance adjusted) estimator was used to estimate the parameters of the EFA models from the correlation matrix (polychoric, polyserial or Pearson correlation). Several EFA models with different number of dimensions (from one to four) have been compared using global fit indexes. The number of dimensions retained was the smallest one with indexes indicating a good fit (parsimony criterion). The global fit indexes used to assess the fit of the EFA models were: Comparative Fit Index (CFI) and Tucker-Lewis Index (TLI) with good fit if >0.95 [24], Root Mean Square Error of Approximation (RMSEA) and its 90% confidence interval (CI) with close fit if <0.05 and fair fit in the 0.05 to 0.08 range [25], Standardized Root Mean square Residual (SRMR) with good fit if <0.08 [26]. To make the dimensions interpretable and delineate the structure, oblique rotations CF-EQUAMAX and CF-FACPARSIM were used in the EFA models for the development of a new structure [27]. After estimation of the number of dimensions, the criteria for generating a structure were the following: (i) a factor loading value greater than 0.4 was retained to assign an item to a dimension [28], and (ii) an item which reflects more than one dimension was assigned to the dimension for which its factor loading was the highest.

**Discriminant validity.** Discriminant validity of the nine BAT-FR ordinal items between groups determined by ECOG scores (0, 1 or 2 vs 3 or 4), global impression of pain control, and changes in treatments' management at inclusion, was assessed by Kruskal-Wallis tests. Differences of items scores between groups of ECOG scores and more differences for breakthrough pain compared to background pain were supposed to show a good discriminant validity [29].

**Convergent and divergent validity.** Convergent validity between BAT-FR and BPI scores was assessed by Spearman correlations between the nine BAT-FR ordinal items scores and BPI items and dimensions scores at inclusion. The higher the number of medium or large correlations, the better the convergent validity is. Correlation coefficient values of > 0.1 were considered as a small correlation, >0.3 a medium correlation and >0.5 a large correlation (in absolute value) [30]. For analgesic treatments at inclusion, convergent validity was assessed by polychoric correlations between BAT-FR ordinal items scores and oral administration of transmucosal fentanyl and divergent validity was assessed by Spearman correlations between BAT-FR ordinal items scores and background medication dosage. These two validities were established if correlations with breakthrough analgesia were higher than correlations with background analgesia [29].

**Test-retest reliability.** The test-retest reliability of each of the nine ordinal item of the BAT-FR was estimated with data at inclusion and visit 2 using weighted Kappa coefficient ($\kappa_w$). Fleiss-Cohen weighting scheme was used. Results' interpretations were: $\kappa_w \geq 0.81$ indicated almost perfect agreement, $0.61 \leq \kappa_w \leq 0.80$ substantial agreement, $0.41 \leq \kappa_w \leq 0.60$ moderate agreement, $0.21 \leq \kappa_w \leq 0.40$ fair agreement, and $\kappa_w \leq 0.20$ slight agreement [31]. The test-retest reliability of the two dimensions (as specified in the English BAT) and the total score of the BAT-FR were estimated using Intraclass Correlation Coefficient (ICC). The results on ICC were interpreted with the previous thresholds as weighted kappa and ICC are equivalent [32].

**Responsiveness.** The responsiveness of the total score of the BAT-FR was estimated using correlation between (a) change of total score of the BAT-FR between visits 1 and 3 and (b) the assessment at visit 3 of Breakthrough Pain (BP) compared with the last week in an ordinal response [29]. Polyserial correlation was used, a correlation greater than 0.5 was considered large [33]. The responsiveness of the two dimensions (as specified in the English BAT) were assessed with the same method (total score of each dimension instead of total score of BAT-FR).

All tests were 2-tailed, and p < 0.05 was considered for statistical significance.

**Software programs.** The CFA and EFA were carried out using Mplus, version 8.4 [22]. All other analyses used SAS software, version 9.4.

**Ethics.** Oral information was delivered to each patient screened for inclusion. If the patient expressed agreement for inclusion, then, the inclusion criteria were verified. If all criteria were met, the patient was given a full oral and written information before signing an informed consent form.

The Committee For Persons' Protection Sud-Est IV was seized on August, 25[th] 2014. It reviewed and approved the protocol on the 10[th] September 2014 (Reference number: L14-155).

## Results

### Participants (Table 1)

One hundred and thirty one participants were included.

The participants were 62 ± 13 years old and diagnosed from cancer for 3.6 ± 3.7 years. Most suffered from gastro-intestinal or urological cancers, which were mainly on a

**Table 1. Population's characteristics.**

| Population characteristics | Number of Patients (N = 130), n (%) | |
|---|---|---|
| Age | Median [Q1; Q3] | 62.2 yrs [53; 70] |
| | Range | 33–100 yrs |
| Delay from the diagnosis of cancer to the completion of the BAT-FR | Median [Q1; Q3] | 2.35 yrs [1.0; 4.9] |
| | Range | 0.03–20.1 yrs |
| Gender | | |
| *Male* | 66 | 50.8% |
| *Female* | 64 | 49.2% |
| Cancer diagnosis | | |
| *Breast* | 13 | 10.0% |
| *Dermatological* | 16 | 12.3% |
| *ENT* | 6 | 4.6% |
| *Gastrointestinal* | 32 | 24.6% |
| *Gynecological* | 13 | 10.0% |
| *Hematological* | 8 | 6.2% |
| *Lung* | 9 | 6.9% |
| *Urological* | 29 | 22.3% |
| *Other* | 4 | 3.1% |
| Stage of cancer | | |
| *Locally advanced* | 17 | 13.1% |
| *Metastatic* | 106 | 81.5% |
| *Recurrence* | 6 | 4.6% |
| *Other* | 1 | 0.8% |
| Cancer treatment | | |
| *Chemotherapy* | 103 | 80.5% |
| *Radiotherapy* | 15 | 11.7% |
| *Surgery* | 8 | 6.3% |
| *Targeted therapy or other* | 2 | 1.6% |
| *Missing* | 2 | |
| ECOG performance status (before the first consultation) | | |
| *0* | 3 | 2.3% |
| *1* | 11 | 8.5% |
| *2* | 60 | 46.2% |
| *3* | 52 | 40.0% |
| *4* | 4 | 3.1% |
| Subject type | | |
| *Outpatient* | 7 | 5.4% |
| *Inpatient* | 123 | 94.6% |
| Pain pathogenesis | | |
| *Cancer-related* | 120 | 92.3% |
| *Cancer treatment-related* | 1 | 0.8% |
| *Mixed* | 9 | 6.9% |
| Pain characteristics | | |
| *Nociceptive* | 46 | 35.4% |
| *Neuropathic* | 6 | 4.6% |
| Mixed | 78 | 60.0% |

metastatic stage. The etiopathogeny of the breakthrough cancer pain was cancer-related for nine in ten participants and had mixed characteristics for two thirds of participants and only nociceptive for around a third of participants. Neuropathic breakthrough cancer pain was minority.

## Questionnaire's acceptability

All items had less than 5% of missing items suggesting a good acceptability, except at visit 3.

## Exploratory factor analysis on the nine ordinal items (Table 2, S2 and S3 Files)

The model with three dimensions had a good fit but unreliable estimates (for both rotations, factor loading >1 and negative variance; bad condition number (for CF-EQUAMAX only) [22]. The two-dimension model was retained because it had a fair fit with reliable parameter estimates (all factor loading <1 and no negative variance, good condition number), and a clear and stable structure whatever the oblique rotation (cf. S4 File for goodness-of-fit and model assumptions of the EFA models). The solution with the oblique rotation CF-FACPARSIM is presented in Table 2 (cf. S3 File for CF-EQUAMAX). The first dimension included five items (2, 6, 7, 8 and 9) and the second dimension included four items (5, 11, 12 and 14). Nevertheless items 2 and 8 had a low contribution to the first dimension and might be excluded (factor loading <0.40) and item 14 was not on the same dimension than for English and Dutch versions (with items 2, 6, 7, 8 and 9). The correlation between the two dimensions was low for the two rotations (r = 0.16 for both).

In the following parts of the article, the two dimensions are, unless otherwise stated, the dimensions based on the structure of the English version.

## Discriminant validity (Table 3)

There was more differences for breakthrough pain than for background pain for items 8, 11 and 12 in control of pain (p-values <0.05) and for items 8 and 12 in changes in management of treatments (p-values <0.05). However, there was more differences for background pain

**Table 2. Items grouping after CF-FACPARSIM rotation: Estimate of the factor loadings associated to each of the two dimensions of the BAT-FR estimated with data of n = 130 patients.**

| French BAT ordinal items | D1 | D2 | Dimension of the original version for this item |
|---|---|---|---|
| How often do you get breakthrough pain? (n˚2) | **0.282** | -0.052 | 1—pain severity and impact |
| How long does a typical episode last? (n˚5) | 0.183 | **0.438***| 2—pain duration and medication |
| How severe is the worst breakthrough pain? (n˚6) | **0.721*** | -0.001 | 1—pain severity and impact |
| How severe is a typical breakthrough pain? (n˚7) | **0.745*** | -0.046 | 1—pain severity and impact |
| How much does the breakthrough pain distress you? (n˚8) | **0.240** | 0.193 | 1—pain severity and impact |
| How much does the breakthrough pain stop you from living a normal life? (n˚9) | **0.549*** | 0.209 | 1—pain severity and impact |
| How effective is the painkiller for your breakthrough pain? (n˚11) | -0.080 | **-0.456*** | 2—pain duration and medication |
| How long does the breakthrough painkiller take to have a meaningful effect? (n˚12) | -0.066 | **0.902*** | 2—pain duration and medication |
| How much do the side effects from your breakthrough painkiller bother you? (n˚14) | 0.008 | **0.481*** | 1—pain severity and impact |

**Bold:** item reflecting more strongly the dimension;

*: factor loading >0.400;

D1: Dimension 1 of the BAT-FR; D2: Dimension 2 of the BAT-FR; 1—pain severity and impact: breakthrough pain severity and impact factor of the English BAT; 2—pain duration and medication: breakthrough pain duration and medication efficacy factor of the English BAT.

**Table 3. Discriminant validity of BAT-FR ordinal items between groups determined by ECOG scores, global impression of pain control and changes in management of treatments, by clinicians at inclusion.**

| | ECOG | | | Control of pain | | | | | | Changes in management of treatments | | | | | |
| --- | --- | --- | --- | --- | --- | --- | --- | --- | --- | --- | --- | --- | --- | --- | --- |
| | | | | Background pain | | | Breakthrough pain | | | Background pain | | | Breakthrough pain | | |
| French BAT ordinal items | 0,1 or 2 (n = 74) | 3 or 4 (n = 56) | p-value | Yes (n = 109) | No (n = 20) | p-value | Yes (n = 68) | No (n = 60) | p-value | Yes (n = 27) | No (n = 102) | p-value | Yes (n = 54) | No (n = 75) | p-value |
| How often do you get breakthrough pain? (n°2) | 6 | 6 | 0.55 | **6** | **8** | **<0.01** | 6 | 6 | 0.42 | 8 | 6 | **0.03** | 6 | 6 | 0.44 |
| How long does a typical episode last? (n°5) | 6 | 6 | 0.87 | 6 | 6 | 0.3 | 6 | 6 | 0.3 | 4 | 6 | 0.3 | 6 | 6 | 0.19 |
| How severe is the worst breakthrough pain? (n°6) | 9 | 8.5 | 0.46 | 9 | 8.5 | 0.55 | 9 | 8.5 | 0.43 | 9 | 9 | 0.47 | 8.5 | 9 | 0.36 |
| How severe is a typical breakthrough pain? (n°7) | 6 | 6 | 0.72 | 6 | 7 | 0.3 | 6 | 6 | 0.37 | 7 | 6 | **0.03** | 6 | 6 | 0.25 |
| How much does the breakthrough pain distress you? (n°8) | 5 | 5 | 0.91 | 5 | 5 | 0.84 | **6** | **4** | **0.03** | 5 | 5 | 0.73 | **4** | **6** | **0.03** |
| How much does the breakthrough pain stop you from living a normal life? (n°9) | 8 | 8 | 0.83 | 7.5 | 9.5 | 0.09 | 8 | 8 | 0.71 | 9 | 8 | 0.23 | 7 | 8 | 0.5 |
| How effective is the painkiller for your breakthrough pain? (n°11) | 7 | 7 | 0.45 | 7 | 5.5 | 0.07 | **8** | **6** | **<0.01** | **5** | **7** | **0.04** | **5** | **8** | **<0.01** |
| How long does the breakthrough painkiller take to have a meaningful effect? (n°12) | 4 | 4 | 0.34 | 4 | 6 | 0.5 | **4** | **5** | **0.03** | 6 | 4 | 0.21 | **5** | **4** | **0.04** |
| How much do the side effects from your breakthrough painkiller bother you? (n°14) | 0 | 0 | 0.71 | 0 | 4 | 0.13 | 0 | 0 | 0.42 | 1 | 0 | 0.14 | 0 | 0 | 0.39 |

Each value in cell is the median of scores by group. **Bold**: item for which p-value<0.05.

than for breakthrough pain for item 2 in control of pain and changes in management of treatments, and for item 7 in changes in management of treatments (p-values <0.05). There were no significant differences of items scores between ECOG groups (all p-values were > 0.34 for the nine items).

## Convergent and divergent validity

Correlations between the BAT-FR and BPI were small to medium (Table 4). Regarding the correlation between the BAT-FR and analgesic treatments (Table 5), the correlation with taking breakthrough analgesia treatment was greater than the correlation with background medication dosage for six items (5, 6, 7, 9, 11 and 14). The difference between coefficients was lower than 0.1 for half items and all correlations were small.

**Table 4. Convergent validity between French BAT and BPI scores at visit 1.**

| French BAT ordinal items | BPI Pain Intensity Items | | | | | | BPI Interference Items | | | | | | | |
|---|---|---|---|---|---|---|---|---|---|---|---|---|---|---|
| | Worst Pain | Least Pain | Average Pain | Pain Now | BPI Pain Intensity Items | % of Pain Relief | General Activity | Mood | Walking | Work | Relations With Others | Sleep | Enjoyment of Life | BPI Interference Items |
| How often do you get breakthrough pain? (n˚2) | 0.09 | 0.07 | 0.20 | 0.09 | 0.18 | -0.07 | 0.18 | 0.06 | *0.31* | 0.19 | 0.07 | 0.00 | 0.11 | 0.18 |
| How long does a typical episode last? (n˚5) | 0.10 | 0.06 | 0.02 | 0.17 | 0.13 | -0.03 | *0.33* | 0.17 | 0.17 | 0.21 | 0.25 | 0.17 | *0.34* | *0.31* |
| How severe is the worst breakthrough pain? (n˚6) | **0.68** | 0.23 | 0.29 | 0.14 | *0.43* | -0.04 | 0.20 | 0.13 | 0.15 | 0.26 | 0.15 | 0.11 | 0.20 | 0.20 |
| How severe is a typical breakthrough pain? (n˚7) | *0.42* | 0.25 | *0.35* | 0.19 | *0.42* | -0.08 | 0.20 | 0.15 | 0.27 | 0.23 | 0.04 | 0.09 | 0.19 | 0.21 |
| How much does the breakthrough pain distress you? (n˚8) | 0.24 | 0.16 | 0.20 | 0.09 | 0.23 | -0.07 | *0.34* | *0.33* | 0.15 | 0.22 | 0.17 | 0.28 | *0.36* | *0.33* |
| How much does the breakthrough pain stop you from living a normal life? (n˚9) | 0.29 | 0.11 | 0.21 | 0.23 | 0.29 | -0.05 | *0.46* | *0.31* | *0.39* | *0.47* | 0.22 | 0.11 | *0.43* | *0.45* |
| How effective is the painkiller for your breakthrough pain? (n˚11) | -0.26 | -0.10 | -0.20 | *-0.33* | *-0.32* | 0.43 | -0.11 | 0.02 | 0.02 | -0.04 | -0.22 | 0.13 | -0.12 | -0.11 |
| How long does the breakthrough painkiller take to have a meaningful effect? (n˚12) | 0.14 | 0.01 | 0.04 | 0.17 | 0.14 | -0.23 | 0.17 | 0.06 | 0.07 | 0.06 | 0.23 | 0.08 | 0.20 | 0.16 |
| How much do the side effects from your breakthrough painkiller bother you? (n˚14) | -0.02 | 0.15 | 0.10 | 0.18 | 0.17 | -0.29 | *0.35* | *0.31* | 0.02 | 0.15 | *0.31* | 0.23 | 0.26 | *0.33* |

BAT = Breakthrough Pain Assessment Tool; BPI = Brief Pain Inventory.

Correlation coefficient values of >0.1 represent a small correlation, >*0.3 medium*, >**0.5 large** (in absolute value).

## Test-retest reliability (Table 6)

All coefficients (weighted kappa and ICC) ranged between 0.31 and 0.66. The reliability was substantial for three items (5, 6 and 8), dimension "breakthrough pain severity and impact", and BAT-FR total score. It was moderate for four items (2, 7, 9 and 12), and dimension "breakthrough pain duration and medication efficacy". Reliability was fair for items 11 and 14.

**Table 5. Convergent validity between the French BAT and breakthrough analgesia and divergent validity between the French BAT and background analgesia at visit 1.**

| French BAT ordinal items | Correlation with taking breakthrough analgesia[a] | Correlation With Background Medication Dosage[b] |
|---|---|---|
| How often do you get breakthrough pain? (n˚2) | -0.08 | 0.17 |
| How long does a typical episode last? (n˚5) | 0.12 | -0.03 |
| How severe is the worst breakthrough pain? (n˚6) | 0.24 | -0.01 |
| How severe is a typical breakthrough pain? (n˚7) | 0.18 | -0.03 |
| How much does the breakthrough pain distress you? (n˚8) | 0.06 | 0.07 |
| How much does the breakthrough pain stop you from living a normal life? (n˚9) | 0.28 | 0.07 |
| How effective is the painkiller for your breakthrough pain? (n˚11) | -0.04 | -0.01 |
| How long does the breakthrough painkiller take to have a meaningful effect? (n˚12) | 0.04 | -0.12 |
| How much do the side effects from your breakthrough painkiller bother you? (n˚14) | -0.04 | 0.03 |

BAT = Breakthrough Pain Assessment Tool;

[a]Transmucosal fentanyl use;

[b]Opioid doses in oral Morphine Equivalent Daily Doses

Correlation coefficient values of >0.1 represent a small correlation, >**0.3 medium**, >**0.5 large** (in absolute value).

## Responsiveness

The assessments of breakthrough pain by patient were available for 102 patients. The polyserial correlations of assessment by patient with change in total score and with change in English-like dimension "breakthrough pain severity and impact" were large (r = 0.59 on n = 89 and n = 95 patients respectively). The polyserial correlation with change in English-like dimension "breakthrough pain duration and medication efficacy" was medium (r = 0.31 on n = 91 patients).

**Table 6. Test-retest reliability of ordinal items, dimensions and total score of the French BAT.**

| Ordinal item, dimension or total score of BAT-FR | Weighted Kappa / ICC | 95% Confidence Interval | n | Interpretation of agreement |
|---|---|---|---|---|
| How often do you get breakthrough pain? (n˚2) | 0.50 | [0.35–0.65] | 129 | moderate |
| How long does a typical episode last? (n˚5) | 0.63 | [0.50–0.76] | 127 | substantial |
| How severe is the worst breakthrough pain? (n˚6) | 0.64 | [0.51–0.78] | 129 | substantial |
| How severe is a typical breakthrough pain? (n˚7) | 0.54 | [0.41–0.68] | 127 | moderate |
| How much does the breakthrough pain distress you? (n˚8) | 0.66 | [0.54–0.78] | 129 | substantial |
| How much does the breakthrough pain stop you from living a normal life? (n˚9) | 0.52 | [0.36–0.67] | 126 | moderate |
| How effective is the painkiller for your breakthrough pain? (n˚11) | 0.31 | [0.09–0.53] | 121 | fair |
| How long does the breakthrough painkiller take to have a meaningful effect? (n˚12) | 0.50 | [0.33–0.68] | 125 | moderate |
| How much do the side effects from your breakthrough painkiller bother you? (n˚14) | 0.40 | [0.21–0.58] | 126 | fair |
| BP severity and impact dimension (English-like dimension) | 0.66 | [0.55–0.75] | 121 | substantial |
| BP pain duration and medication efficacy dimension (English-like dimension) | 0.55 | [0.41–0.66] | 118 | moderate |
| BAT total score | 0.65 | [0.53–0.74] | 114 | substantial |

# Discussion

## Main findings

This study aimed to assess the psychometric properties of a translation of the BAT in French language and in particular its validity. By comparing with the results of the original BAT [13], items 2 (frequency) and 8 (distress) should be excluded from the items of the BAT-FR, and item 14 (side effects) would change dimension. However, the structure for the BAT-FR has two dimensions like the original one; with for each dimension at least 3 items in common with the corresponding dimension of the original BAT. We therefore consider the dimensions of the BAT-FR as close to those of the original version. However, the values of the factor loadings are mostly lower for the BAT-FR compared to the English BAT. Note that, for the Dutch version of the BAT [14], the structure is identical to the English BAT but with factor loadings of generally lower values as well, therefore the results of the exploratory factor analysis on the data of the BAT-FR are not in contradiction with the results of the English BAT.

It is therefore necessary to decide on keeping, and the dimensions to which to assign items 2 (frequency), 8 (distress) and 14 (side effects) in the BAT-FR. This must be done taking into account the clinical relevance, the meaning of the two dimensions for BAT-FR and for English BAT and also the results of the other aspects of the validation: convergent validity with BPI (poor for the item 2, moderate for items 8 and 14), convergent and divergent validities with pain treatments (poor for items 2, 8 and 14), discriminant validity (poor for items 2 and 14, moderate for item 8), reliability (moderate for item 2, substantial for item 8, fair for item 14).

First, decision must be made on items 2 (frequency) and 8 (distress). Both items measures key concepts in pain assessment that are clinically highly relevant for clinicians. For item 2, it can reflect the equilibration of the background pain treatment and guide tailored treatment adaptation [34, 35]. While item 8 reflects the global impact of pain on patients. Inadequately controlled pain leads to anxiety, depression and sleep disorders [36]. Assessing the distress associated with the experience of pain might trigger non-pharmacological interventions for managing it, such as psychosocial support, resulting in better quality of life. We therefore, would suggest keeping these items into the BAT-FR by assigning them the dimension on which the factor loading is the greatest. The "pain severity and impact" English BAT dimension would then be chosen making the structure of the BAT-FR even closer to that of the original version.

Second, for item 14 (side effects), the values of the factor loadings clearly indicate a change in dimension (from the English BAT dimension "pain severity and impact" to "pain duration and medication"). Both dimensions are relevant for this item. In the original English version, it might be argued that medication side effects are part of the patient's burden related to pain and contributes to its impact which explains that both contributes to the same dimension [37]. However, the treatment side effects are key information to make a choice on whether or not pursing the medication. It has a strong impact on medication adherence [38, 39], and it might lead to treatment discontinuation or adaptation [40]. Therefore, we would argue that item 14 is clinically more relevant to be included in dimension "pain duration and medication", despite the fact that it differs for the original version.

The differences observed between the two tools might be related to the sample involved. When compared to the sample used by Webber et al. for the initial instrument validation, both studies had mainly gastro-intestinal cancers and urological cancers, suggesting that these cancers might be more prone to trigger breakthrough cancer pain. However, in our study, the majority of included patients had impaired Performance Status when the majority of patients included in Webber et al. had 1 to 2 ECOG scores. Our population was also mainly concerned by mixed cancer pain when there were a minority compared to nociceptive pain in Webber

et al. study. These differences in our population might influence the experience of pain [41, 42], resulting in the observed differences.

### Strengths and weaknesses

The sample was composed exclusively of cancer patients. In order to validate the BAT-FR, it should undergo other validation assessments in another sample and in non-malignant conditions.

### Conclusion

The BAT-FR had a close structure with the original version except for the item 14 related to side effects. All properties (validity, reliability and responsiveness) have moderate quality, but are relatively consistent with those of the original version. The BAT-FR needs assessment on another sample to confirm the structure and its properties.

### Supporting information

**S1 File. French version of the Breakthrough Pain Assessment Tool (OFEA).**
(DOCX)

**S2 File. Indexes of fit for exploratory factorial analysis models with one to four dimensions estimated with data of n = 130 patients.** CFI: comparative fit index; TLI: Tucker-Lewis Index; RMSEA: root mean square error of approximation; CI: Confidence interval; SRMR: Standardized Root Mean Square Residual.
(DOCX)

**S3 File. Items grouping after CF-EQUAMAX rotation: Estimate of the factor loadings associated to each of the two dimensions of the French BAT estimated with data of n = 130 patients.**
(DOCX)

**S4 File. Table reporting the goodness-of-fit and model assumptions of the EFA models.** The condition number is an index of good quality of numerical results (ratio of smallest to largest eigenvalue for the Information Matrix, good if $>10^{-6}$) e.g. no problem as multicolinearity of items.
(DOCX)

**S5 File. Database supporting the study results.**
(PDF)

### Acknowledgments

We acknowledge all participants who agreed to take part to the study.

The authors sorely Mrs Dr. Catherine Mercier, PhD, who passed away and has extensively, contributed to this work. Beyond this study, Catherine Mercier has greatly contributed to improve the quality of biostatistics in research to always purse the goal of a better and trustworthy research in health.

### Author Contributions

**Conceptualization:** Elise Perceau-Chambard, Sylvain Roche, Colombe Tricou, Catherine Mercier, Cécile Barbaret, Marilène Filbet.

**Data curation:** Guillaume Pierre Serge Economos.

**Formal analysis:** Sylvain Roche, Catherine Mercier.

**Funding acquisition:** Elise Perceau-Chambard.

**Investigation:** Elise Perceau-Chambard, Colombe Tricou, Cécile Barbaret, Guillaume Pierre Serge Economos.

**Methodology:** Catherine Mercier, Andrew Davies, Katherine Webber, Marilène Filbet, Guillaume Pierre Serge Economos.

**Project administration:** Elise Perceau-Chambard.

**Resources:** Andrew Davies, Katherine Webber, Guillaume Pierre Serge Economos.

**Supervision:** Elise Perceau-Chambard, Colombe Tricou, Andrew Davies, Marilène Filbet.

**Validation:** Colombe Tricou, Guillaume Pierre Serge Economos.

**Writing – original draft:** Guillaume Pierre Serge Economos.

**Writing – review & editing:** Elise Perceau-Chambard, Sylvain Roche, Colombe Tricou, Cécile Barbaret, Andrew Davies, Katherine Webber, Marilène Filbet.

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
