## [Decision Letter · Decision Letter 0]

12 Jan 2023

PONE-D-22-26784Validation of a French version of the Breakthrough Pain Assessment Tool in cancer patients: Factorial structure, reliability and responsiveness.PLOS ONE

Dear Dr. Guillaume,

Thank you for submitting your manuscript to PLOS ONE. After careful consideration, we feel that it has merit but does not fully meet PLOS ONE’s publication criteria as it currently stands. Therefore, we invite you to submit a revised version of the manuscript that addresses the points raised during the review process.

We look forward to receiving your revised manuscript.

Kind regards,

Supat Chupradit, Ph.D., M.Ed., B.Sc.(OT), B.P.A., B.Ed., B.A.

Academic Editor

PLOS ONE

Journal Requirements:

Reviewers' comments:

Reviewer's Responses to Questions

**Comments to the Author**

1. Is the manuscript technically sound, and do the data support the conclusions?

Reviewer #1: Yes

Reviewer #2: Yes

Reviewer #3: Partly

2. Has the statistical analysis been performed appropriately and rigorously? 

Reviewer #1: No

Reviewer #2: Yes

Reviewer #3: No

3. Have the authors made all data underlying the findings in their manuscript fully available?

Reviewer #1: Yes

Reviewer #2: Yes

Reviewer #3: Yes

4. Is the manuscript presented in an intelligible fashion and written in standard English?

Reviewer #1: No

Reviewer #2: Yes

Reviewer #3: Yes

5. Review Comments to the Author

Reviewer #1: 1:The methods and results in the abstract section need to be fundamentally rewritten

2: In the last paragraph of the introduction: did you aim to develop the French version or to examine its psychometrics؟

3:Please give a complete explanation of the main questionnaire. For example, how many questions does it have? How is the scoring method?

4:Please clearly explain who, how, and when permission from the questionnaire designer (who is?) was obtained.

5:Which face and content validity methods have been used?

6:How did you diagnose cognitive impairment?

7:Please explain the Brief Pain Inventory (BPI).

8:Please explain ECOG.

9:How did you calculate internal consistency?

Reviewer #2: Overall, the article is interesting. The process to validate the French version of the Breakthrough Pain Assessment Tool is well-explained. I only have a few suggestions to improve the quality of the article.

• Since it was stated that “1268 patients were screened for inclusion. One-hundred and thirty-one were included and received a questionnaire”, it is necessary to clarify the procedure used to retain only 131 patients (and exclude more than 1,137 patients from the study).

• The sample size is important when running the factor analysis (either EFA or CFA); In this case, it was vital to ensure that the total number of 130 patients was adequate for statistical analysis (The KMO and Bartlett test may be necessary to further analyze if the sample size used in the manuscript was adequate.).

• The result of the EFA analysis of the BAT-FR yielded 11 items, while the original English version has 14 items. We wondered why the BAT-FR contained a number of items that were different from the original after reconsidering items #2, 8 and 14 (all of which remained in the BAT-FR). Perhaps it would be necessary to have an in-depth discussion on this matter.

Reviewer #3: 1-The authors should consider in assumption check for using EFA e.g. vif, correlation levels etc.

2-The authors should exmine the language validity by used the correlation between language expert.

3-The discriminant analysis should compare between model, because this study examine both EFA and CFA.

6. PLOS authors have the option to publish the peer review history of their article (what does this mean?). If published, this will include your full peer review and any attached files.

Reviewer #1: No

Reviewer #2: No

Reviewer #3: No

---

## [Author Response · Author response to Decision Letter 0]

31 Mar 2023

Dear Editor,

We appreciate the Academic Editor’s consideration of our contribution to Plos One. 

Please find bellow our answers to the reviewer’s comments:

Answers to Reviewer #1 comments

1) The methods and results in the abstract section need to be fundamentally rewritten:

We have rewritten the abstract. 

2) In the last paragraph of the introduction: did you aim to develop the French version or to examine its psychometrics?

It is a very fair comment and we would like to thank the reviewer for this. We aimed to translate it in French language and to examine its psychometric; I modified the text according to it.

“Our study aimed to translate the BAT into French and assess the psychometric properties (validity, reliability and responsiveness) of a French version of the Breakthrough Pain Assessment Tool (BAT-FR).”

3) Please give a complete explanation of the main questionnaire. For example, how many questions does it have? How is the scoring method?

To address this requirement, we modified the Material and Methods section to include the following paragraph: 

“a- The Breakthrough Pain Assessment Tool (BAT):

The BAT is a 14 question tool designed to clinically assess breakthrough pain in cancer patients.(13) The tool includes nine questions related to pain and five related to its management. The first question uses a body shape to locate the painful areas. The next questions uses free text (four questions), 10-points rating scales (six questions), and categorical scales (three questions).”

4) Please clearly explain who, how, and when permission from the questionnaire designer (who is?) was obtained.

Prof. Andrew Davies and Dr. Katherine Webber are the original designers of this tool. Their permission has been obtained before the validation. Actually, this project was co-developed with them. They are listed as authors of the manuscript. We added te following sentence to the methods section: 

“The overall process was co-supervised by the original developers of the instrument who are co-authors of this article.”

5) Which face and content validity methods have been used?

These properties are usually assessed by qualitative methods. The translation and adaptation process maintains these two properties as for the English version. We added the following sentence in the part about Translation and adaptation of the original tool: “This entire process with experts and a small sample of patients gave evidence for good content and face validity.”

6) How did you diagnose cognitive impairment?

Cognitive impairment was considered as an exclusion criteria when it was listed in the medical history of the patient.

We modified the following sentence to be more explicit : 

“Non-inclusion criteria were already known cognitive impairment (when listed in the medical history of the patient) and overwhelming fatigue.”

7) Please explain the Brief Pain Inventory (BPI).

We thank the reviewer for this requirement.

We added the references related to this instrument and the following paragraph: 

“The BPI is an instrument designed to assess the severity of pain and the functional impairment due to pain. The instrument includes one categorical scale question, one body shape for the localization of pain, four 10-points Likert-scales for rating pain intensity at various times, one free-text question and seven numerical scales to assess the impact of pain on the patient’s daily life.”

8) Please explain ECOG.

We thank the reviewer for asking this additional information. As ECOG is widely known from the clinical community, we only added a reference and a short explanation as follows: 

“Performance Status rated using the five-point Eastern Cooperative Oncology Group (ECOG) tool,(17)”

9) How did you calculate internal consistency?

Cronbach’s alpha is the coefficient traditionally used to assess internal consistency of a dimension but its assumptions (continuous score for each item, linearity between the underlying dimension and item score, equality of factor loadings of items of a same dimension, non-correlation between the residuals) are rarely met (Bollen, K. A. (1989). Structural Equations with Latent Variables. Wiley; Vet, H. C. W. de (Éd.). (2011). Measurement in medicine : A practical guide. Cambridge University Press; Yang, Y., & Green, S. B. (2011). Coefficient alpha : A reliability coefficient for the 21st century? Journal of Psychoeducational Assessment, 29(4), 377‑392) and Cronbach’s alpha underestimates internal consistency (Green, S. B., & Yang, Y. (2009). Reliability of Summed Item Scores Using Structural Equation Modeling : An Alternative to Coefficient Alpha. Psychometrika, 74(1), 155‑167.). As the present results clearly do not meet these assumptions, Cronbach’s alpha was not estimated. Moreover, to estimate a reliability coefficient is relevant after CFA, not EFA. Other coefficients as Raykov’s rho, Revelle’s beta or Green and Yang coefficient (Yang, Y., & Green, S. B. (2015). Evaluation of Structural Equation Modeling Estimates of Reliability for Scales with Ordered Categorical Items. Methodology, 11(1), 23‑34) will be relevant to estimate the internal consistency once the structure of the BAT-FR will be confirmed in a future study.

 

Answers to Reviewer #2 comments

1) Since it was stated that “1268 patients were screened for inclusion. One-hundred and thirty-one were included and received a questionnaire”, it is necessary to clarify the procedure used to retain only 131 patients (and exclude more than 1,137 patients from the study).

We thank the reviewer for pointing this important lack of clarity.

I omitted to report the exclusion criteria and number of patients excluded for each cause. To address this issue, I modified the paragraph to the following one:

“Exclusion criteria were the absence of pain (597), refusing to take part (20), agony (207), tiredness (13), inability to consent (235), trouble in understanding French (14), having already participated (23). Seven were excluded for an unknown reason. It resulted in one-hundred and thirty one included patients who received a questionnaire.”

2) The sample size is important when running the factor analysis (either EFA or CFA); In this case, it was vital to ensure that the total number of 130 patients was adequate for statistical analysis (The KMO and Bartlett test may be necessary to further analyze if the sample size used in the manuscript was adequate.).

We add in Material and Methods the following paragraph for the sample size:

“As for the English version, the nine ordinal items were used to obtain scores and the five nominal items gave additional information and help to interpret the scores. To determine the sample size for factor analysis, we used the rule of 5 x number of parameters.(20) This rule gave 130 patients because the model of the confirmatory factor analysis on the nine ordinal had 26 parameters (with the same structure as the English structure and items with 11 response categories considered as continuous). Moreover, this sample size was greater than the number of parameters for the exploratory factor analysis (up to 36 for four dimensions). According to COSMIN guidelines, a sample size of 130 was considered adequate to assess the other properties of the questionnaire.(21)”

We added the following paragraph in the part about Factorial structure of the BAT-FR in the statistical aspects:

“To assess if factor analysis is relevant, a test similar to the Bartlett's sphericity test was used: the Chi-Square Test of Model Fit for the Baseline Model of uncorrelated dependent variables, available in Mplus (Muthén & Muthén,2017) for model fit information (p-value ≤ 0.05 indicates that a factor analysis may be useful with the data).”

Moreover, this test is relevant if some item scores are ordinal. In the Additional file 4, we added a table of model fit of the different EFA models with the p-value of this test, the global fit indexes and the number of parameters of the models. We have also modified in the results of exploratory factor analysis, the following sentences (modifications in bold):

“The model with three dimensions had a good fit but unreliable estimates (for both rotations, factor loading >1 and negative variance; bad condition number (Muthén & Muthén,2017) for CF-EQUAMAX only). The two-dimension model was retained because it had a fair fit with reliable parameter estimates (all factor loading <1 and no negative variance, good condition number), and a clear and stable structure whatever the oblique rotation (cf. Additional file 4 for goodness-of-fit and model assumptions of the EFA models).”

3) The result of the EFA analysis of the BAT-FR yielded 11 items, while the original English version has 14 items. We wondered why the BAT-FR contained a number of items that were different from the original after reconsidering items #2, 8 and 14 (all of which remained in the BAT-FR). Perhaps it would be necessary to have an in-depth discussion on this matter.

For original English version, factor analysis was on the 9 ordinal items: n°2, 5, 6, 7, 8, 9, 11, 12 and 14 (see Weber 2014, table 3). The 5 other items were nominal (free text or no order in the response categories) and thus could not be subject to a factor analysis. 

For the French version, CFA and EFA were also on these 9 ordinal items. The 5 other items were nominal, so they could not be in EFA or CFA; they gave additional information for the clinicians to interpret the ordinal items. 

We added in the abstract, the statistics aspects, the results, the tables and the discussion about the structure of the BAT-FR that the factor analyses are on the nine ordinal items. We also added the following sentences at the beginning of the part about Factorial structure of the BAT-FR in the statistical aspects:

“As for the original English version, factor analysis was on the nine ordinal items: n°2, 5, 6, 7, 8, 9, 11, 12 and 14 (see Weber 2014, table 3). The five other items were nominal (free text or no order in the response categories) and thus could not be subject to a factor analysis; they gave additional information for the clinicians to interpret the ordinal items.”

Answers to Reviewer #3 comments

1) The authors should consider in assumption check for using EFA e.g. VIF, correlation levels etc.

We are in the framework of latent variable models and check the model fit and assumptions with usual index and tests for these models (CFI, TLI, RMSEA, SRMR, Chi-Square Test of Model Fit for the Baseline Model…) all implemented in Mplus. VIF is used to detect multicolinearity in usual regression models. In the current framework, multicolinearity of items leads to singular information matrix and the condition number of the information matrix (ratio of its smallest to its largest eigenvalue) must be greater than 10-6 to ensure good numerical results (Muthén, L.K. and Muthén, B.O. (1998-2017). Mplus User’s Guide. Eighth Edition. Los Angeles, CA: Muthén & Muthén). 

We added in the table of Additional file 4 for goodness-of-fit and model assumptions a column with the condition number of the EFA models and a note to explain the condition number. 

2) The authors should examine the language validity by used the correlation between language expert.

We thank the reviewer for this comment.

As described in the method section, during the translation and adaptation process of the original tool, we strictly followed the steps of the guidelines for cross cultural adaptation process. The first step relies on bilingual translators whose mother language is the targeted language. The translation from English to French was performed by a French native, then, the back-translation from French to English was performed by a native English speaker. Then, an expert committee met to discuss the concepts being assessed. This committee included one language specialist. The other members were methodologists and health care professionals. The tool was finally tested as a pre-final version in the targeted population. As the initial development team was part of the process they were contacted for expert advises all along the process. 

3) The discriminant analysis should compare between models, because this study examine both EFA and CFA.

The discriminant validity was assessed at the level of the nine ordinal items of the BAT-FR (cf. Table 3), not at the level of dimensions of the EFA and CFA models. Thus, no comparison between models was done. 

We hope that the amended manuscript is now suitable to be published in PlosOne and remains available for any additional comment,

Best regards, 

 Drs. Sylvain Roche and Guillaume Econom

---

## [Decision Letter · Decision Letter 1]

30 May 2023

Validation of a French version of the Breakthrough Pain Assessment Tool in cancer patients: Factorial structure, reliability and responsiveness.

PONE-D-22-26784R1

Dear Dr. Guillaume,

We’re pleased to inform you that your manuscript has been judged scientifically suitable for publication and will be formally accepted for publication once it meets all outstanding technical requirements.

Kind regards,

Iftikhar Ahmed Khan

Academic Editor

PLOS ONE

Additional Editor Comments (optional):

Reviewers' comments:

Reviewer's Responses to Questions

**Comments to the Author**

1. If the authors have adequately addressed your comments raised in a previous round of review and you feel that this manuscript is now acceptable for publication, you may indicate that here to bypass the “Comments to the Author” section, enter your conflict of interest statement in the “Confidential to Editor” section, and submit your "Accept" recommendation.

Reviewer #1: All comments have been addressed

Reviewer #2: (No Response)

Reviewer #3: All comments have been addressed

2. Is the manuscript technically sound, and do the data support the conclusions?

Reviewer #1: Yes

Reviewer #2: Yes

Reviewer #3: Yes

3. Has the statistical analysis been performed appropriately and rigorously? 

Reviewer #1: Yes

Reviewer #2: Yes

Reviewer #3: Yes

4. Have the authors made all data underlying the findings in their manuscript fully available?

Reviewer #1: Yes

Reviewer #2: (No Response)

Reviewer #3: Yes

5. Is the manuscript presented in an intelligible fashion and written in standard English?

Reviewer #1: Yes

Reviewer #2: Yes

Reviewer #3: Yes

6. Review Comments to the Author

Reviewer #1: Validation of a French version of the Breakthrough Pain Assessment Tool in cancer patients: Factorial structure, reliability and responsiveness=Accept

Reviewer #2: (No Response)

Reviewer #3: 1-Due to used the rule of thumb for setting the sample size that lead to the effect size and power of test in this study.

2-The another evidence that support the discriminant validity is comparative between model fit indicate between sub group.

well revise version

7. PLOS authors have the option to publish the peer review history of their article (what does this mean?). If published, this will include your full peer review and any attached files.

Reviewer #1: No

Reviewer #2: No

Reviewer #3: No

---

## [Editor Report · Acceptance letter]

28 Jun 2023

PONE-D-22-26784R1 

Validation of a French version of the Breakthrough Pain Assessment Tool in cancer patients: Factorial structure, reliability and responsiveness. 

Dear Dr. Serge Economos:

I'm pleased to inform you that your manuscript has been deemed suitable for publication in PLOS ONE. Congratulations! Your manuscript is now with our production department. 

Kind regards, 

on behalf of

Dr. Iftikhar Ahmed Khan 

Academic Editor

PLOS ONE